# Physical Activity in Eating Disorders: A Systematic Review

**DOI:** 10.3390/nu12010183

**Published:** 2020-01-09

**Authors:** Rizk Melissa, Mattar Lama, Kern Laurence, Berthoz Sylvie, Duclos Jeanne, Viltart Odile, Godart Nathalie

**Affiliations:** 1INSERM U1178, Maison de Solenn, 97 Boulevard De Port Royal, 75014 Paris, France; Nathalie.godart@fsef.net; 2Université Paris-Sud and Université Paris Descartes, Ecole Doctorale des 3C (Cerveau, Cognition, Comportement), UMR-S0669, 75006 Paris, France; 3Psychiatry Unit, Institut Mutualiste Montsouris 42, Boulevard Jourdan, 75014 Paris, France; Sylvie.berthoz-landron@inserm.fr; 4Nutrition Program, Department of Natural Sciences, Lebanese American University, Beirut 1102, Lebanon; lama.mattar@lau.edu.lb; 5Laboratoire EA 29 31, LINP2-APSA, et Laboratoire EA 4430 CLIPSYD Université Paris Nanterre UFR-STAPS, 200, Avenue de la République, 92001 Nanterre CEDEX, France; 6INCIA UMR-5287 CNRS, Université de Bordeaux, 33076 Bordeaux, France; 7Sciences Cognitives et Sciences Affectives, Université de Lille, CNRS, UMR 9193—SCALab, 59045 Lille, France; jeanne.duclos@univ-lille.fr; 8Département de Psychiatrie, Hôpital Saint Vincent de Paul, GHICL, F-59000 Lille, France; 9Institute of Psychiatry and Neurosciences of Paris, Unité Mixte de Recherche en Santé (UMRS) 1266 Institut National de la Santé et de la Recherche Médicale (INSERM), University Paris Descartes, 75014 Paris, France; odile.viltart@inserm.fr; 10Department of Biology, University of Lille, 59000 Lille, France

**Keywords:** review, eating disorders, physical activity, problematic use of physical activity

## Abstract

Abnormally high levels of physical activity have been documented throughout the literature in patients with eating disorders (ED), especially those diagnosed with anorexia nervosa (AN). Yet no clear definition, conceptualization, or treatment of the problematic use of physical activity (PPA) in ED patients exists. The aim of this review is to propose a new classification of PPA, report the prevalence, triggers, predictors, maintainers and other related factors of PPA in ED patients, in addition to proposing a comprehensive model of the development of PPA in AN. A total of 47 articles, retrieved from Medline and Web of Science, met the inclusion criteria and were included in the analysis. As a result, the new approach of PPA was divided into two groups (group 1 and group 2) according to the dimension (quantitative vs qualitative approach) of physical activity that was evaluated. The prevalence of PPA in ED was reported in 20 out of 47 studies, the comparison of PPA between ED versus controls in 21 articles, and the links between PPA and psychological factors in ED in 26 articles, including depression (16/26), anxiety (13/26), obsessive–compulsiveness (9/26), self-esteem (4/26), addictiveness (1/26), regulation and verbal expression of emotions (1/26) and anhedonia (1/26). The links between PPA and ED symptomatology, PPA and weight, body mass index (BMI) and body composition in ED, PPA and age, onset, illness duration and lifetime activity status in ED, PPA and ED treatment outcome were reported in 18, 15, 7, 5 articles, respectively. All of the factors have been systematically clustered into group 1 and group 2. Results focused more on AN rather than BN due to the limited studies on the latter. Additionally, a model for the development of PPA in AN patients was proposed, encompassing five periods evolving into three clinical stages. Thus, two very opposite components of PPA in AN were suggested: voluntarily PPA increased in AN was viewed as a conscious strategy to maximize weight loss, while involuntarily PPA increased proportionally with weight-loss, indicating that exercise might be under the control of a subconscious biological drive and involuntary cognition.

## 1. Introduction

A display of abnormally high levels of physical activity has been observed from the earliest clinical description of eating disorders (ED), especially in anorexia nervosa (AN) [1]. The latter has been considered to affect 31% to 80% of AN patients [2] and has been associated with a longer length of hospital stay [3], poor treatment outcome [4], interfering with refeeding strategies and body weight stabilization [5] and an increased risk of relapse and chronicity [6]. With more than 400 articles and seven partial reviews published on the subject in the past three decades, there is still no consensus on how to define, conceptualize or treat these observed high levels of physical activity in individuals suffering from ED. Physical activity is considered to be any body movement produced by the contraction of skeletal muscles, resulting in a substantial increase of energy expenditure relative to basal metabolism [7]. This has translated into a plethora of terms and definitions that has been described in social psychology by Hagger [8] who defines it as a “déjà-vu” phenomenon: “the feeling that one has seen a variable with the same definition and content before only referred to by a different term” [8] (p. 1). This might imply inconsistent or even contradictory findings, when in fact the definitions are the ambiguous factor [8]. In the current review, problematic use of physical activity in ED will be referred to as “problematic use of physical activity or PPA”.

The seven previous reviews tackling the matter [2,9,10,11,12,13,14] did not adhere to the recommendation suggested by Hagger [15]. In other terms, they did not propose a redefinition/a model of the particular phenomena of PPA in ED. None of them considered all currently published studies, as some have been published earlier and/or have focused on only one aspect/dimension of the topic.

Researchers have given a plethora of terms and definitions to describe PPA in eating disorder patients [11]. Terms included “hyperactivity”, “compulsive exercise”, “driven exercise”, “unhealthy exercise”, “motor restlessness”, “over-exercise”, “overactivity”, “hard exercise”, “drive to exercise”, “drive for activity”, or “exercise dependence”. This inconsistent use of terminology points out the ambiguity in defining this problematic behavior [16]. As mentioned by Adkins and Keel [17], definitions generally include a quantitative dimension of the behavior, including volume, frequency and/or intensity of exercise and/or a psychopathological dimension, while mentioning compulsivity and/or obsession and/or dependence to exercise. Multiple possible amalgamations can, therefore, be present and confuse the readers [15].

In light of the current gaps in the literature, the aims of this review are to:–Propose a new classification of PPA based on the characteristics of physical activity and the nature of the relation that links the individual to his/her physical activity, independently of the name the authors had originally given it.–Give an overview of the prevalence of PPA in ED patients; as well as the triggering, predicting, maintaining and other associated factors, using our proposed new classification of PPA, which takes into account the reported methodological discrepancies.–Propose a comprehensive model of the development of PPA in AN.

## 2. Materials and Methods

A systematic literature search was performed according to the Preferred Reporting Items for Systematic Reviews and Meta-Analysis (PRISMA) statement [18].

### 2.1. Data Sources and Search Strategies

The information and relevant studies were retrieved by searching two on-line databases, MEDLINE and Web of Science. The following search strategy was conducted (adapted for each database): (“Eating disorders” OR “anorexia nervosa” OR “bulimia nervosa”) AND (“Exercise” OR “physical activity”). Only articles in English or French, from all countries, published between the years 1970 and September 2019 (included) were reviewed. To identify further potentially relevant studies for inclusion, a complementary manual search of the titles, abstracts and reference lists of these articles was performed. When data were missing in the paper, additional information from study authors was systematically sought.

### 2.2. Study Selection

Eligible studies in the empirical literature should have aimed to evaluate the prevalence, the frequency, the nature and/or the clinical associated features (psychological or somatic) of physical activity in AN, Bulimia Nervosa (BN) and Eating Disorders Not Otherwise Specified (EDNOS). Our inclusion criteria limited our review to human studies that recruited all participants, or at least part of them, diagnosed with AN, BN with or without the mention of the subtypes; or EDNOS according to research diagnostic criteria. Thus, all studies on participants from the general population, athletes or animal models were excluded. Studies including other disorders such as Binge Eating Disorder or Attention Deficit Hyperactivity Disorder and case reports were excluded. All studies that did not match the aim of our review or papers on the treatment and/or management of physical activity in ED were not considered. The seven reviews mentioned in the introduction were also excluded as previously mentioned: none of them considered all currently published studies, as some are old and/or focused on only one aspect of the topic. Two independent reviewers (MR and NG) performed eligibility assessment individually. Disagreements between reviewers were resolved by consensus. Finally, 47 studies that met the inclusion criteria were identified and included in this paper. A flowchart summarizing the flow diagram of study selection is presented in Figure 1.

It was not possible to conduct a meta-analysis of the available studies due to the diversity of concepts and definitions used to evaluate PPA in ED.

## 3. Results

Aim number 1: Propose a new classification of PPA based on the characteristics of physical activity and the nature of the relation that links the individual to his/her physical activity, independently of the name the authors had originally given it.

Table 1 presents a plethora of terms and definitions used by authors, as well as the instruments used to assess PPA and the time of assessment.

We propose a new approach of PPA. It will categorize and differentiate the quantitative and the qualitative conceptualisations used in the literature, independently of the terms used by the authors, and divide studies into the following two groups:–Group 1: Studies classified in this group determined PPA in a quantitative approach of physical activity in terms of intensity, frequency, duration and/or type of physical activity.–Group 2: Studies classified in this group determined PPA in a qualitative approach by focusing on the relation that links an individual to his/her physical activity, including motives for exercise, compulsivity and exercise dependence/addiction, with or without a quantitative measure of physical activity.

Group 1 included 20 studies, group 2 included 20 studies, and seven studies were simultaneously in group 1 and group 2 (Table 2, column 2).

Table 1 summarizes terms used by authors, definitions, cut-offs, assessment instruments and time of assessment of PPA in ED. Results of study designs, diagnostic criteria of ED and sample composition are detailed in Table 3.

Aim number 2: Give an overview of the prevalence of PPA in ED patients and of the triggering, predicting, maintaining and other associated factors, using our proposed new classification of PPA, which takes into account the reported methodological discrepancies.

### 3.1. Prevalence of Problematic Use of Physical Activity in ED (20/47 Studies)

#### 3.1.1. Problematic Use of Physical Activity in AN (20/20) and the Subtypes

In Group 1, the current prevalence of PPA varies from 30% to 75% according to the definitions or cut-off chosen (nine studies; Table 2, column 3).

In Group 2, we found 14 prevalence for PPA in 12 studies. Lifetime prevalence were given by four studies [49,59,61,66]. Current prevalence varied from 34% to 58% (Table 2, column 3). The two with the largest sample size [63,65] were both in favor of higher PPA in AN-R patients compared to AN-BP. Klein et al. [57], who compared only 8 AN-R with 13 AN-BP patients, failed to find a significant difference.

Shroff et al. [61], who described lifetime ED subtypes (usually diagnosed transversely) found significantly higher PPA in ANP (according to their own categorization, cf. Table 3 for details of categorization) than all other subtypes of AN including, what they called, restrictive AN and BN participants and individuals with a lifetime diagnosis of AN and BN.

#### 3.1.2. Problematic Use of Physical Activity in BN (9/20)

Among BN patients, both prevalence of PPA in Groups 1 and 2 seemed to be strongly affected by the way they were defined and measured. There were no studies comparing BN patients to healthy controls.

In Group 1 (2/9), PPA varies between 32.6% and 39% in two studies [3,49] (Table 2, column 4).

In Group 2 (1/9), regarding lifetime prevalence, the only study available reported 57% of PPA in BN [48]. Five studies considered current BN status and reported the prevalence of PPA as varying from 20.2% (either current or past ED; [61]) to 60% [69].

#### 3.1.3. Comparison of Problematic Use of Physical Activity between AN and BN (10/20)

In the studies comparing PPA in AN and BN [3,41,44,48,49,54,61,63,64,83], AN were found to be physically more active than BN in: (a) Group 1, four studies [3,41,61,83] out of five [49] found no significant difference for current exercise status. (b) Group 2, three out of six studies [44,48,63] while the three others found no significant difference [49] (only for lifetime exercise status) [54,83].

### 3.2. Comparison of Problematic Use of Physical Activity between ED Versus Controls (21/47)

#### 3.2.1. Problematic Use of Physical Activity in AN Patients Versus Controls (21/21)

Prevalence of PPA in Group 1 was investigated in two studies [58,76], and the results were contradictive. Davis et al. [58] found significantly more excessive exercisers among the AN patients than among healthy controls. However, physical activity was significantly higher in healthy controls compared to AN inpatients [76].

Energy expenditure was measured in six studies [40,41,52,70,75,80] and exercise duration in one study [83]. Energy expended during physical activity [40,70] and duration of exercise [83] were significantly higher in AN patients compared to control subjects.

#### 3.2.2. Problematic Use of Physical Activity in BN Patients Versus Controls (2/21)

In Group 1, no significant group differences were found both in energy expenditure (in kilo calories per day) [41], nor in the duration of physical activity (in minutes per day) [41,83].

In Group 2, BN patients scored higher on the CET’s rule-driven behaviour, weight control exercise, mood improvement and exercise rigidity subscales than healthy controls [83].

### 3.3. Links between Problematic Use of Physical Activity and Comorbidities and Psychological Factors in ED (26/47)

The comorbidities and psychological factors that have been investigated in ED patients will be reviewed below in the following order: depression (16/26), anxiety (13/26), obsessive–compulsiveness (9/26), self-esteem (4/26), addictiveness (1/26), regulation and verbal expression of emotions (1/26) and anhedonia (1/26) (Table 4). When BN is not mentioned, the latter implies absence or non-significant data.

#### 3.3.1. Depression in AN

Group 1 included six studies and the majority of them (5/6) did not find any association between PPA and depression [62,67,73,75,76]. Only Falk et Al. [38], found that patients that were less active at admission were also more depressed, but this association disappeared after 14 days of hospitalization.

Group 2 included eight studies: five [54,56,65,74,75] reported a positive proportional association between depression, on both categorical and dimensional approaches, and exercise status, including frequency and volume of exercise. Furthermore, Zipfel et al. [74] found that the patient’s level of depressive symptoms was positively correlated to total daily energy expenditure. In addition, out of the four studies that found no association between depression and PPA, Long et al. [42] found nonetheless that depression was associated with a greater tendency for secret and solitary exercising in anorexics. The two remaining studies were those with the smallest samples (*n* = 30 and *n* = 21) [56,57].

#### 3.3.2. Anxiety in AN

Group 1 included four studies [62,67,73,75], which all failed to find an effect of anxiety: (a) when comparing high/non-high exercisers [62,73]; (b) using correlations between the level of anxiety and general physical activity measured in counts/day by an accelerometer [67] or actimetry [75]; (c) when comparing two levels of physical activity that are sedentary and light physical activity [67]. However, Carrera et al. [67] noticed that the more time AN patients spent in the moderate to vigorous physical activity level, the less they reported to be anxious (cf. Table 4).

Group 2 included nine studies. All studies except one [65] found an increase in PPA in cases of elevated anxiety.

Furthermore, Long et al. [42] found that anxiety levels and phobic anxiety were associated with a greater tendency for solitary and secret exercising in AN. They also found that individuals suffering from AN were more likely to cope with negative emotional states (referring to feelings of anxiety, anger, low mood/depression) by using exercise than normal controls.

In addition, Penas-Lledo et al. [54] found that AN patients who exercised had significantly greater levels of somatization in addition to anxiety than the ones who did not exercise. Thornton et al. [66] found that significantly more individuals suffering from AN with or without generalized anxiety disorder (GAD) reported more PPA than healthy controls and women with GAD only.

#### 3.3.3. Obsessive–Compulsiveness in AN

Group 1 included only two studies [73,78], which did not find a significant difference between excessive and non-excessive exercisers. However, Blachno et al. [78] found that adolescent patients at high risk of obsessive–compulsive disorder reported more intentional physical activities aimed at weight loss than the non-high-risk group.

Group 2 included seven studies: Obsessive–compulsiveness was found to be associated with PPA in three of these studies [45,50,59,81], with pathologically motivated exercisers reporting more obsessive–compulsive personality characteristics and greater obsessional-compulsive disorder symptoms than non-pathologically motivated exercisers [59]. Surprisingly, Bewell-Weiss and Carter [65], who used a totally different instrument (Padua Inventory, [88]) found that obsessive–compulsiveness was negatively associated with PPA in AN. The remaining two studies [54,56] found no association between obsessive–compulsiveness and PPA. No statistical association was found between obsessive–compulsive traits and addictive personality in childhood and adolescent activity levels of AN patients [50].

#### 3.3.4. Self-Esteem and Addictiveness in AN

Self-esteem was found to be positively associated with PPA in patients suffering from AN in Group 2 [65,81]. The study by Davis et al. [50], which is also classified into Group 2, found that an addictive personality in AN significantly predicted the degree to which patients reported obligatory attitudes to exercising. There were no data in the literature regarding BN patients.

#### 3.3.5. Regulation and Verbal Expression of Emotions in BN

A study by Brownstone et al. [72], classified into Group 2, found that only BN patients with high levels of compulsivity engaged in more PPA when they presented high levels of affect lability compared to low levels of affect lability. They found a trend for the same association in the case of high compulsivity’s association between PPA and restricted emotion.

#### 3.3.6. Anhedonia in ED

The study by Davis and Woodside [53], which can be classified into Group 1, found that exercise status, i.e., excessive vs. non-excessive exercisers, contributed positively to the variance in the level of anhedonia reported by patients with AN, independently of patient differences in depression severity. They also found that BN patients belonging to the non-excessive exercise group had significantly lower anhedonia scores than both BN patients in the excessive exercise group and AN patients belonging to the non-excessive exerciser group.

### 3.4. Links between Problematic Use of Physical Activity and ED Symptomatology in AN (18/47)

ED symptomatology was studied either globally or focused on one aspect, such as weight preoccupation, drive for thinness, body dissatisfaction or eating behaviors.

Group 1 included eight studies. Three examined general eating disorder pathology and found no association with either locomotor activity [62], actimetry [75], or levels of physical activity [73,76]. When taking into account peak activity (highest per minute level of activity in a 5 min period), PPA was correlated to EDE-Q higher scores [75].

However, significant associations were reported when the core ED symptoms were studied separately. In fact, Davis et al. [45] found that weight preoccupation was positively related to AN patients’ level of physical activity. Two studies [40,51] focused on comparing ED symptoms between AN patients and healthy controls and found that PPA in AN patients was significantly associated with greater body dissatisfaction and drive for thinness than controls. Nevertheless, Carrera et al. [67] found that body dissatisfaction subscale scores of the Eating Disorders Inventory (2nd version) were not a significant predictor of PPA in AN.

Group 2 included seven studies with mainly the following results: (1) Weight preoccupation was found positively related to PPA among AN patients [58]; (2) Greater levels of general eating symptoms and bulimic scores were found in AN outpatients who exercised than in those who did not [54]; (3) Drive for thinness was found to be positively associated with PPA in AN [74]; (4) Quantitative food restriction was found to be positively related to PPA in AN patients, even when anxiety was taken into account [56]; (5). PPA was positively correlated with general eating pathology scores of AN outpatients [75,77] and AN inpatients [81]. In fact, Keyes et al. [75] found that ED pathology, along with exercising to improve mood, contributed the most to the variance in PPA in these patients (25.3% and 26.5%, respectively).

### 3.5. Links between Problematic Use of Physical Activity and Weight, Body Mass Index (BMI) and Body Composition in ED (15/47)

Five studies used bioelectrical impedance to assess body composition in their study samples, [40,41,73,76,82]; yet none used a validated equation for ED population [89]. Given the differences in instruments used to measure PPA, very heterogeneous results were observed.

#### 3.5.1. Weight, BMI and Body Composition in AN (12/12)

Group 1 included 11 studies [38,40,41,46,57,67,73,76,79,80,82]. Four studies failed to find a significant difference between excessive and non-excessive exercisers or physical activity levels with regard to body mass index (BMI) [41,57,67,73,80] and body composition [41,73]. Other studies found that body weight [38,82], BMI [40,46,76,82], resumption of menses [70] and percentage of body fat [40,82] were positively related to the level of daily physical activity. Moreover, Bouten et al. [46] found that the intensity of physical activity (in counts per minute, measured by an accelerometer) in AN outpatients increased significantly with BMI values ≥ 15 kg·m^−2^. Below this value, physical activity was assumed to reach a minimum.

Group 2 included four studies with different results. Two studies [56,77] found that BMI was not significantly correlated to PPA. However, Keyes et al. [75] found that PPA was negatively correlated to BMI. Penas-Lledo et al. [54], in a clinical chart analysis, found that AN patients who reported PPA had higher BMIs than the ones who did not.

#### 3.5.2. Weight, BMI and Body Composition in BN (2/11)

In Group 1, Pirke et al. [41] found no association between PPA, BMI or body composition. In Group 2, Penas-Lledo et al. [54] found that BN patients who reported PPA had higher BMIs than non-exercisers.

### 3.6. Links between Problematic Use of Physical Activity, Age, Onset, Illness Duration, Quality of Life and Lifetime Activity Status in ED (7/49)

#### 3.6.1. Age, Onset, Illness Duration and Lifetime Activity Status in AN (7/7)

Group 1 included five studies. The link between patient’s age and PPA was examined in three studies, with concordant results. None of the studies found an association between AN inpatient’s age and their level of locomotor activity [62] or of physical activity [67,73].

Davis et al. [58] measured physical activity levels during childhood and found an increase of the level of physical activity with age for both AN patients and controls; however, a greater rate was observed for AN patients. In addition, a large change in the amount of physical activity approximately one year prior to the onset of AN was detected.

Moreover, Davis et al. [48] found that AN patients who exercise excessively had a significant increase in physical activity levels around one year prior to the onset of the disease. An earlier age of onset of AN was associated with both quantitative and qualitative dimensions of PPA [48,73].

Finally, Kostrzewa et al. [73] found that, before admission to treatment, AN excessive exercisers had longer illness duration than the non-excessive exercisers group.

Group 2 included three studies that measured lifetime PPA status among AN adolescents. Davis et al. [48] found that the currently “more active” patients were also more involved in sport and exercise activities at ages 10 and 12 than their less-active counterparts. Davis et al. [50] found that a greater number of patients who had been highly active as children subsequently engaged in more PPA during their illness than those who were average/less active as children. However, Noetel et al. [81] failed to find an association between PPA and duration of illness in their adolescent sample.

#### 3.6.2. Age, Onset, Illness Duration and Lifetime Activity Status in BN (1/7)

The study by Davis et al. [48], which can be classified into Group 2, found that a greater proportion of BN adult patients started dieting before starting to regularly exercise (or dieted without ever exercising) than the ones who started exercising before reducing their food consumption.

#### 3.6.3. Quality of Life (2/7)

Quality of life has been found to be negatively impacted by the effect of PPA (in Group 2) interacting with ED severity [84,90].

### 3.7. Links between Problematic Use of Physical Activity and ED Treatment Outcome (5/49)

#### 3.7.1. ED Treatment Outcome in AN (4/5)

Group 1 included five studies. Kostrzewa et al. [73] found that recovery, defined as having a score equal or superior to nine on the Morgan and Russell outcome assessment schedule, was predicted by significantly higher physical activity levels at inclusion. They also found a decrease in activity levels for excessive exercisers at one year of follow-up; however, the activity levels stabilized after that. However, Lehmann et al. [82] did not find intensity and duration of physical activity to be significant predictors of percentage of BMI increase between admission and discharge to an inpatient treatment. Gianini et al. [80] found total physical activity to be higher in patients at a weight restoration of 90% in an inpatient treatment compared to the start of hospitalization.

El Ghoch et al. [70] studied treatment dropouts and their results were twofold: (1) They found a positive association between PPA and treatment dropout in AN inpatients, with dropouts having higher moderate and vigorous physical activity duration and expenditure at baseline than completers. They failed to find an association between end-of-treatment physical activity assessment measures ED or general psychiatry features between AN completers vs. dropouts. They also failed to find a significant difference in EE assessment when comparing AN subtypes of patients who completed their treatment with those who did not. (2). For the comparison between AN patients who completed their treatment with healthy controls, AN completers were found to have a higher number of daily steps, duration and moderate–vigorous physical activity expenditure. Kostrzewa et al. [73] did not confirm this result in their adolescent (and smaller) sample.

El Ghoch et al. [79] found that the number of daily steps at inpatient discharge was the only independent predictor of menstrual resumption; the non-menstruating inpatient group performed a higher number of daily steps than the menstruating group at discharge.

#### 3.7.2. ED Treatment Outcome in BN (1/5)

A study by Smith et al. [69], which can be included in Group 2, were the only ones to examine, in a sample of adult suffering from BN, the links between PPA and suicide in a multivariate analysis. Smith et al. [69] found that PPA was positively associated with suicidal gestures and attempts even when controlling for frequency of vomiting episodes, laxative abuse, dietary restraint and age.

## 4. Discussion

We hypothesized that categorizing studies into two groups would lead to a harmonization of their results and make a comparison between them possible (cf. results aim 1). This was only partially verified. Indeed, the classification we proposed did not decrease the wide range of prevalence generally given in the literature of 31% to 80% [2]. Our group classification showed that a high level of PPA (group 2) is associated with more anxiety, obsessive–compulsiveness and addictiveness, and higher self-esteem, which was not the case for studies using PPA evaluations (group 1). We were also able to confirm that AN patients have more PPA than healthy controls independently of the quantitative or qualitative dimension studied.

### 4.1. Prevalence of Problematic Use of Physical Activity

The ranges of prevalence in both cases are inarguably highly dependent on: (1) the cut-off used by authors to divide their study sample into high vs. low exercisers, sometimes doubling from one study to another; (2) the sample composition (inpatients vs. outpatients and AN vs. BN); (3) evaluation methods (subjective vs. objective measures). Indeed, current prevalence was considered during hospitalization, which directly or indirectly limits PPA [5]. Furthermore, methods of self-evaluation are highly critical among AN patients who are known to hide/underestimate (considering self-reports) their symptoms [91].

In addition, an important factor affecting PPA prevalence was that lifetime PPA was nearly two-times higher than the current prevalence (80% vs. 46%); therefore, we considered those data separately.

### 4.2. Comparison of Problematic Use of Physical Activity between ED Patients Versus Controls

Compared to healthy controls, AN patients had higher frequencies of PPA if PA was measured subjectively. Objectively, healthy controls were found to be more physically active than AN inpatients [76]. This is not surprising since, as mentioned previously, hospitalized patients have limited PA.

A lack of research devoted to AN patients that have low to no physical activity levels in comparison with those having high PPA was observed. We are compelled to think that they could (1) have low BMI since Bouten et al. [46] found that physical activity reached a minimum at BMI values below 15 kg·m^−2^; and/or (2) be more depressed, as Falk et al. [38]; and/or (3) more frequently classified as AN binge–purging type [44].

At least partially against this view, Davis [9] discussed that AN patients feel an increasingly strong compulsion to be physically active, despite pain and exhaustion. No studies focus on the question of the threshold of nutritional statuses that lead to a decrease in PPA, nor if this absence of adaptive and progressive decrease in PPA, linked to exhaustion, could define a subgroup of AN patients, observed clinically. This subgroup is the one that could have PPA until death.

### 4.3. Problematic Use of Physical Activity, Comorbidities and Psychological Factors

Since depression, anxiety and obsessive–compulsive disorder frequently co-occur with AN [92,93,94], it was not surprising to find that they were the most studied psychological factors to be linked together with ED and PPA. We can see here all the interest of using our classification to re-analyze inconclusive results found in previous reviews, with results from Group 1 frequently being the opposite than Group 2.

PPA only considered quantitatively (Group 1) was found to be negatively correlated to depression [38] or completely not associated to it. This was also the case in the general population, where greater amounts of occupational and leisure time activities, as well as all physical activity intensity levels, were generally associated with reduced symptoms of depression [95,96], anxiety and stress [38].

However, when focusing on PPA (group 2) and thus taking into account the pathological motivations linked to ED, we observed the opposite effect: depression (or depressive disorder), anxiety, obsessive–compulsiveness, addictiveness, and higher self-esteem were significantly positively associated with PPA. The more an AN patient has those symptoms, the more he/she practiced PPA. This could be explained by the fact patients were trying to improve their mood and emotional state [9,42,54,75,83] with physical activity being considered as a way of coping with a chronically negative affect [37] by pursuing the boosting effect of endogenous opioids [10].

### 4.4. Problematic Use of Physical Activity and ED Symptomatology

Keyes et al. [75] found that ED pathology was one of the most important predictors of the variance of PPA in Group 2 in AN outpatients, but when compared to healthy controls, PPA in AN patients were significantly and positively associated with drive for thinness. It seems that AN patients are likely to increase their levels of PPA (quantitatively and qualitatively) when preoccupied about their weight and motivated by their drive for thinness. Vansteelandt el al. [37] explained high levels of physical activity in ED throughout psychological mechanisms and emphasized the fact that AN and BN patients have place too much value in their body shape and weight. Thus, physical activity is a result of a conscious attempt to work-off calories and obtain their desired thinness ideal, making the drive for thinness an important motive to engage in PPA. This is also the case in the general population, where weight and shape control reasons for exercise participation are very common [97].

### 4.5. Problematic Use of Physical Activity and Weight, BMI and Body Composition

Findings should be interpreted with caution due to the absence of validated methods to assess body composition in ED patients [98]. In both our classification and independently of ED types (AN and BN), almost half the studies did not find an association between different intensities of physical activity and BMI or body composition. The other half found a significantly positive association between PPA and body weight, BMI or percentage of body fat. It is in accordance with the findings of Bouten et al. [46] who showed that there was a threshold of BMI at 17 kg·m^−2^ among adult AN outpatients under which patients decreased their physical activity. They tried explaining these decreases in PPA and BMI with a decrease in muscle mass, diminished muscular function, malnutrition and exhaustion [46]. This goes in agreement with the findings of [39], who suggested that an increased physical activity occurs in AN patients during weight gain and recovery periods, as well as during the treatment phase and restoration of body fat [73].

### 4.6. Problematic Use of Physical Activity Link with Age of Onset, Age and Lifetime Activity Status in AN

Very few studies adjusted their results to age. In Group 1, only the larger study over two studies [61,67] found that PPA was associated with a younger age at time of interview but this was in a sample of ED where AN and BN patients where not distinguished [61]. Two explanations were proposed for this result: (1) the easy access to exercise behaviors as a weight-control mechanism at younger ages, rather than having access to other purging behaviors such as laxatives and purgatives; (2) the existence of an inverse relationship between exercise and age, which is also found in the general population, where physical activity tends to decrease with age [99].

In addition, PPA in Group 2 was related to physical activity during childhood: Davis et al. [48,50] found that AN patients who had been highly active as children were found to be engaged in more PPA during their illness. These findings suggest that high levels of physical activity during childhood could predict the development of PPA in AN, implying possible individual profile variations independently of the disorder. This goes in agreement with findings from the general population implying that sports participation during childhood and adolescence is particularly predictive of being more physically active later in life [99].

### 4.7. Problematic Use of Physical Activity and AN Treatment Outcome

The contradictory results between two studies [70,73] that, respectively, included adults and adolescents, raise the question of the contribution of age to the results. Indeed, there is a lower frequency of dropout in teens [100] combined with the lowest severity of ED at adolescence [101], and higher physical activity during childhood and adolescence than among young adults and older age groups [102].

### 4.8. Problematic Use of Physical Activity and Compulsivity

Continuing normal to high activity levels despite weight loss and a negative energy balance is recognized as an exclusive specificity of AN patients compared to individuals in a situation of starvation due to other causes such as those implicating an increase in circulating inflammatory cytokines [103] in humans as well in animal models [104,105].

In animal models mimicking several symptoms of AN in an attempt to give a rather biological approach, a reduction in food intake and body weight was paradoxically accompanied by a progressive increased activity level. The most described animal model combining food restriction and voluntary physical activity is the “Activity-Based Anorexia” model (ABA model) developed in the rat by Routtenberg and Kuznesof [106], then in the mouse. This model is also called “starvation-induced hyperactivity” [107] or “semi-starvation induced hyperactivity” [108]. In fact, the rodents have free access to a running wheel and showed hyperactivity occurring in response to a limited food supply due to limited access time to food (1–2 h per day). Such behavior, occurring at 2–3 days after the beginning of the protocol, led to feedback inhibition of food intake or self-starvation and death in 5–6 days [109]. Thus, the negative energy balance state that becomes life threatening, eventually leads to death [2,110]. A lot of research has been done on animal models to try understanding and determining the biological phenomenon underneath this particular hyperactivity, which occurred several hours before food distribution and is called food anticipatory activity. Similar activity has been also described in AN patients [111]. The animal models will not be detailed as it is beyond the scope of this systematic review.

### 4.9. A Proposed Comprehensive Model of the Development of PPA in AN

At this point, we can propose, based on literature reviews and clinical practice, a comprehensive model of the development of PPA in AN (Figure 2), taking into account both the history of the patient, his/her interaction with his/her environment and the pathological consequences of AN. This model is divided into five periods: Period 0 entitled “factors preceding AN”, Period 1 entitled “onset of AN”, Period 2 entitled “evolution of AN”, Period 3 entitled “acute phase of AN” and Period 4 entitled “long-term outcome”. In parallel, these periods evolve in three clinical phases (number 1, 2 and 3), with voluntary and involuntary components varying with time.–In Period 0: The main points that should be taken into account are a patient’s childhood activity profile [48], having a more physically active father [58], participation in esthetic- or weight-oriented sports [112]. An increase in PPA is observed one year prior to the onset of the disease [48].–In Period 1: An increased PPA is majored with an early age of onset [61]; PPA as a conscious strategy for AN patients to optimize weight loss, also found in the general population [9,97].

Clinical phase number 1: Patients program a physical activity determined in defined moments of the day. This physical activity progressively increases in volume, intensity and/or frequency. PPA is described by patients as voluntary [10] and “goal-directed, organized and planned” [23].–In Period 2: PPA becomes a coping strategy to compensate, suppress, and/or alleviate both negative affective states (anxiety [54,56], depression [54] and stress [20,75]) and ED symptoms [75] including, weight preoccupation [45,58], drive for thinness [40,51,74] body dissatisfaction [40] and restrictive profile [63,65].–This is in addition modulated by ambient temperature [67].

Clinical phase number 2: As the ED progresses, physical activity could become an increasingly autonomous process. Involuntary PPA appears, with automation of the behavior. For example: patients run instead of walking, stand up when they normally should be sitting down, for example while writing or eating, or walk instead of standing still. This PPA is associated with diffuse restlessness and a significant unsteadiness, where the patient is literally unable to stand still for a short period. At a certain point, patients will even maintain muscular tension such as keeping their gluteal or abdominal muscles contracted without even thinking about it. The latter is called “static PPA”. Furthermore, patients will continue to try to maximize their daily energy expenditure by all voluntary means possible. For example, they walk to get from one place to another instead of using a car, use stairs instead of elevators, do more housework, etc.–In period 3: PPA has now a composite nature. It has a compulsive component not under voluntary cognitive control of the patient [9], irrespective of the antecedent motivation to exercise. Nevertheless, patients also present voluntary PPA due to a current context (for example: increased activity motivated by body dissatisfaction and weight preoccupation such as weight increase). The respective proportions of voluntary vs. involuntary PPA vary in a given subject both according to patients themselves and according to time. Clinical practice suggests, as it has been observed in biological animal models that a small subgroup of patients could (or can) be hyperactive until death with a huge hyperactivity despite very low weight.

Clinical phase number 3: PPA is described by patients as “more intense, driven, disorganized and aimless than it was” [23] and as “aimless, stereotyped and inefficient” [10]. This phase associates clinical manifestations number 1 and 2, with varying degrees. We can notice here three profiles of patients: (1) hypo- or normal-active patients; (2) patients who relatively control important PPA; (3) patients who can’t control their extremely important and solitary PPA (sometimes until death), associated unsteadiness and static physical activity.–In period 4: A Long-term outcome is not very well known. Long-term weight recovered patients have been found to have a frequency of PPA similar to healthy controls [39]. Furthermore, PPA seemed to be associated with more dropout [70] and could be a negative factor for the extremely active patients.

### 4.10. Limitations

It is important to acknowledge the limitations of our work. Firstly, the term “problematic use of physical activity” is a term we have decided to use. Unfortunately, it does add to the plethora of terms already found in the litterature. This highlights the urgent need for the development of an international and interdisciplinary task force that would need to find and vote on a commonly accepted term and definition. Secondly, some studies [45,49,50,52,74,75,83] were added in both groups 1 and 2 because they evaluated PPA both quantitatively and qualitatively. Despite the fact that this had been taken into account while analysing the results of these specific studies, it can still be considered as an external bias. Finally, our model is an attemp to synthetise the elements of the literature, the clinical elements we collected from persons with AN, in order to better understand PPA. All individuals with AN do not have PPA or exactly the same PPA. The component we described can be present at the same time or at different times. We hope that this model will help explain the complexity of the phenomen and help treat it better.

## 5. Conclusions

Having no consensus definition for what is PPA in ED, the interpretation of previous studies findings can be partially jeopardized. In order to address this problem, we attempt to propose in our review a definition of this clinical concept including quantitative (intensity, frequency, duration and type of physical activity) and qualitative (motives for exercise, compulsiveness and exercise dependence/addiction) elements only in AN, given that very little information was found on BN. Thus, this paper highlights two main opposite component of PPA in AN: (a) PPA that is sometimes voluntarily increased in AN for the soul purpose of burning-off more calories and thus viewed as a conscious strategy of AN patients to optimize weight loss; (b) PPA that sometimes is involuntarily increased simultaneously with weight-loss and low ambient temperatures; indicating that exercise might not be under voluntary cognitive control of the patient, with a subconscious biological drive, a part of this activity becoming totally automatic. Additionally, according to the literature and clinical experience, a chronological model for the development of PPA in AN patient was proposed encompassing five periods evolving into three clinical stages.

In conclusion, we hypothesized that the evaluation of the intensity, frequency, duration and type of physical activity along with the motives for exercise, compulsiveness and exercise dependence for each person with ED would allow more individualized and efficient medical therapies.

Future research initiatives should focus on finding and voting on a commonly accepted term and definition for PPA in ED.

## Figures and Tables

**Figure 1 nutrients-12-00183-f001:**
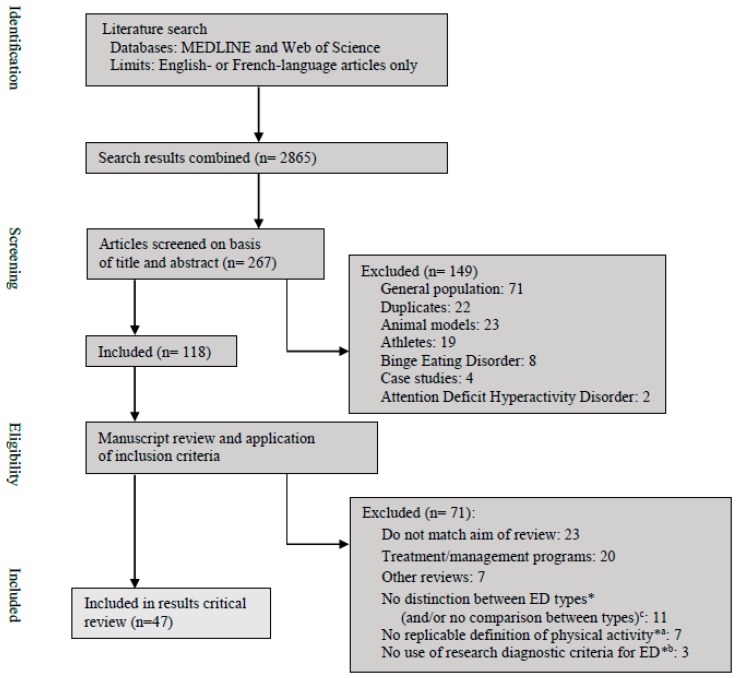
Flow diagram of the study selection. * Studies excluded for one or more reasons. ^a^ Blumenthal et al. (1984) [19]; Crisp et al. (1980) [20]; Davis et al. (1994) [21]; Frey et al. (2000) [22]; Kron et al. (1978) [23]; Monell et al. (2018) [24]; Sharp et al. (1994) [25]. ^b^ Blumenthal et al. (1984) [19]; Higgins et al. (2013) [26]; Long and Hollin (1995) [27]. ^c^ Boyd et al. (2007) [28]; Bratland-Sanda et al. (2010a, 2010b and 2011) [29,30,31]; Carruth and Skinner (2000) [32]; Davis et al. (1994 and 1998) [21,33]; Hechler et al. (2008) [34]; Naylor et al. (2011) [35]; Stiles-Shields et al. (2011) [36]; Vansteelandt et al. (2007) [37]. ED—eating disorders.

**Figure 2 nutrients-12-00183-f002:**
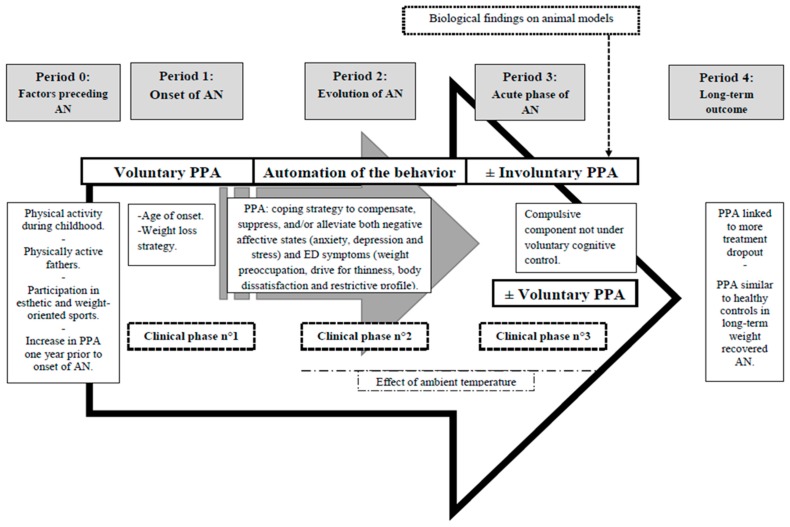
Proposed comprehensive model of development of problematic use of physical activity (PPA) in AN.

**Table 1 nutrients-12-00183-t001:** Terms used by authors, definitions, cut-offs, assessment instruments and time of assessment of problematic use of physical activity in ED (columns 1 to 5).

1	2	3	4		5
References Country	Terms Used by Authors	Definitions and Cut-Offs	Assessment Instrument		Time of Assessment
			Objectively	Subjectively	
Falk et al. (1985) [38] USA	Energy expenditure/motor activity	-	Actimetry	-	First 2 weeks of hospitalization.
Kaye et al. (1986) [39] USA	Physical activity	-	Acc.	-	3 to 5 day period of stable weight in hospital.
Casper et al. (1991) [40] USA	Energy expenditure	-	DLW	-	At time of admission to treatment program.
Pirke et al. (1991) [41] Germany	Hyperactivity and energy expenditure	-	DLW	PA diary	2 weeks after consenting to study (3 to 6 weeks in hospital).
Long et al. (1993) [42] Canada	Hyperactivity	-	-	1. ORQ 2. Modified CRQ (Carmack and Martens, 1979 [43])	First 2 weeks of hosp. in an ED unit.
Brewerton et al. (1995) [44] USA	Compulsive exercise	Exercise to control weight at least once a day and exercised for at least 60 min	-	DSED	-
Davis et al. (1995) [45] Canada	Excessive exercise	Exercise on average, a minimum of 5 h per week during the year prior to assessment	-	1. Interview 2. PA diary 3. CES	4 weeks and year prior to study inclusion.
Bouten et al. (1996) [46] The Netherlands	PA levels	Three PA levels: Low, moderate or high level.	1. Acc. 2. DLW	-	7 days after study inclusion.
Casper and Jabine (1996) [47] USA	Excessive exercise	Exercise more than 4 h per week of exercise for the month preceding the intake.	-	Interview	Month prior to hospitalization.
Davis et al. (1997) [48] Canada	Excessive exercise	Level of PA considerably more than typical for someone their age at that time and time spent exercising exceeded 1 h per day for at least 6 days per week for a period not less than 1 month, and if she described the exercising as “obsessive,” “driven,” and “out of control” during the excessive phase.	-	1. Interview 2. CES	Minimum 1 month prior to study inclusion.
Davis and Claridge (1998) [49] Canada	Excessive exercise	Lifetime exercise status: idem as Davis et al. (1997) [48]. Current exercise status: exercise activity for a minimum of 6 h a week averaged over the 12 months prior to assessment.	-	Interview	Year prior to study inclusion and during lifetime.
Davis et al. (1999) [50] Canada	Excessive exercise	Current exercise status: exercise activity for a minimum of 6 h a week averaged over 1 month prior to assessment.	-	1. Interview 2. CES	Minimum 1 month prior to study inclusion.
Favaro et al. (2000) [51] Italy	Excessive exercise	Excessive exercise as at least 1 h of intensive physical activity per day.	-	Interview	At start of outpatient treatment.
Pinkston et al. (2001) [52] USA	Energy expenditure	Three activity levels: Moderate, hard or very hard activities during the past 7 days.	-	1. 7d PAR 2. CES	At hospitalization.
Solenberger (2001) [3] USA	PA levels	“Patients were categorized into high- or low-level exercise groups by a median split of total exercise. The high-level exercise group spent greater than 6.7 h/week exercising”.	-	Clinical charts	Six months prior to hospitalization.
Davis and Woodside (2002) [53] Canada	Excessive exercise	Exercise activity for a minimum of 6 h a week averaged over 1 month prior to assessment.	-	Interview	Minimum 1 month prior to study inclusion.
Penas-Lledo et al. (2002) [54] Spain	Excessive exercise	Physical exercise at least 5 times a week, for at least 1h without stopping, and with the aim of burning calories.	-	Clinical charts	-
Holtkamp et al. (2003) [55] Germany	Motor restlessness	Five levels of PA: no excessive physical activity; slight and/or rare excessive physical activity; marked and/or occasional excessive physical activity; strong and/or frequent excessive physical activity; very strong excessive physical activity.	-	SIAB-EX	3 months prior to hospitalization.
Holtkamp et al. (2004) [56] Germany	Excessive physical activity	Idem as Holtkamp et al. (2003) [55]	-	SIAB	3 months prior to hospitalization.
Klein et al. (2004) [57] USA	Exercise dependence	Exercise dependence if greater frequency of exercise ≥3 criteria of Modified SDSS	-	1. SDSS 2. CES	4 weeks prior to hospitalization.
Davis et al. (2005) [58] Canada	Excessive exercise	“Considerably more exercise/physical activity than is typical or normal for someone your age, and beyond the requirements of any competitive sport in which you were engaged”.	-	1. Interview 2. CES	Year prior to hospitalization and lifetime.
Davis and Kaptein (2006) [59] Canada	Excessive exercise	Idem as Davis and Claridge (1998) [49] for lifetime and current exercise statuses.	-	Interview	Year prior to study inclusion and during lifetime.
Holtkamp et al. (2006) [60] Germany	PA levels and motor and inner restlessness	Idem as Holtkamp et al. (2003) [55]	Acc.	1. SIAB-EX 2. Self-report measures	3 months prior to hospitalization.
Shroff et al. (2006) [61] USA	Excessive exercise	Excessive exercise when any of the following were reported by the participant: (1) severe interference with important activities; (2) exercising more than 3 h/day and distress if unable to exercise; (3) frequent exercise at inappropriate times and places and little or no attempt to suppress the behavior; and (4) exercising despite more serious injury, illness or medical complication.	-	SIAB	3 months prior to hospitalization.
Klein et al. (2007) [62] USA	Excessive exercise	Exercising at least 6 h per week, on average.	Acc.	1. Interview 2. CES	3 months prior to hospitalization and during first 2 weeks of hospitalization.
Dalle Grave et al. (2008) [63] Italy	Compulsive exercise	Compulsive exerciser in the presence of a positive answer to the first question below and to anyone of the remaining: (1) Over the past 4 weeks, have you exercised with the aim of burning up calories to control your shape or weight? (2) Have you felt compelled or obliged to exercise? (3) Have you exercised even when it caused severe interference with important activities? (4) Have you exercised to a level that might be harmful for you? (5) Have you felt distressed if you were unable to exercise?	-	EDE	4 weeks prior to hospitalization.
Mond and Calogero (2009) [64] Australia	Excessive exercise	Exercise “hard as a means of controlling their shape or weight during the preceding 4 weeks”.	-	1. EDE-Q 2. CES 3. REI	4 weeks prior to hospitalization.
Bewell-Weiss and Carter (2010) [65] Canada	Excessive exercise	Minimum of 1 h of obligatory exercise aimed at controlling shape and weight, 6 days per week in the month before admission.	-	EDE	3 months prior to hospitalization.
Thornton et al. (2011) [66] Sweden and USA	Excessive exercise	Exercise more than 2 h per day to control her shape and weight.	-	Interview	At time of interview.
Carrera et al. (2012) [67] The Netherlands	Physical activity levels/hyperactivity	Three PA intensities: Sedentary (<200 counts/min), light (200 to <1800 counts/min) and moderate to vigorous (≥1800 counts/min) activity.	Acc.	-	During 3 days after study inclusion.
Murray et al. (2012) [68] Australia	Compulsive exercise	-	-	CET	4 weeks prior to hospitalization.
Smith et al. (2012) [69] USA	Over-exercise	Exercise “hard as a means of controlling their shape or weight during the preceding 4 weeks”.	-	EDE-Q	4 weeks prior to hospitalization.
El Ghoch et al. (2013) [70] Italy	PA energy expenditure	Two PA levels: light PA (<3 METs) and moderate and vigorous PA: (≥3 METs)	Acc.	-	During 1st week of hosp. and last week of day hospital.
Alberti et al. (2013) [71] Italy	PA energy expenditure	Three PA levels: Low PA (<3 METs), moderate PA (3 to 6 METs) and vigorous PA (>6 METs).	1. Acc. 2. IPAQ	-	At hospitalization.
Brownstone et al. (2013) [72] USA	Hard exercise	Exercise “hard as a means of controlling their shape or weight during the preceding 4 weeks”.	-	EDE-Q	4 weeks prior to study inclusion.
Kostrzewa et al. (2013) [73] The Netherlands	PA levels and moderate-to-vigorous PA	Three levels of PA: Sedentary (<200 counts/min), light (200 to <1800 counts/min) and moderate-to-vigorous physical (≥1800 counts/min) activity.	Acc.	-	At hosp., end of hosp. and at 1 year follow-up.
Zipfel et al. (2013) [74] Australia	PA	From EDI-SC: “What percentage of your exercise is aimed at controlling your weight >50% cutoff”	DLW	1. DDE 2. EDI-SC	During hosp.
Keyes et al. (2015) [75] United Kingdom	Drive to exercise	-	Actimetry	1. CES 2. OEQ GDES 3. EAI 4. IPAQ REI	During 7 days after study inclusion.
Sauchelli et al. (2015) [76] Spain	PA	Two PA levels: low exercisers (<300 min spent in moderate-to-vigorous PA) and high exercisers (≥300 min spent in moderate-to-vigorous PA).	Acc.	-	During 6 days after study inclusion.
Sternheim et al. (2015) [77] The Nertherlands	Drive for activity	“urge to be physically active and an inability to sit still”	-	DFA-Q	At time of admission to treatment program.
Blachno et al. (2016) [78] Poland	PA	-	-	PAQ	Week prior to study inclusion.
El Ghoch et al. (2016) [79] Italy	PA energy expenditure	Two PA levels: light PA (<3 METs) and moderate and vigorous PA: (≥3 METs).	Acc.	-	During last week of day hospital.
Gianini et al. (2016) [80] USA	PA	-	Acc.	-	At hosp., at 90% weight gain in hospitalization and at 1 month after hospital discharge.
Noetel et al. (2016) [81] Australia	Compulsive exercise	-	-	CET	4 weeks prior to hospitalization.
Lehmann et al. (2018) [82] Germany	PA energy expenditure	Four PA levels: Very light-intensity PA ([1.1; 1.8] METs), light-intensity PA ([1.8; 3]), moderate-intensity PA ([3; 6] METs) and vigorous-intensity PA (≥6 METs).	Acc.	-	During hospitalization.
Schlegl et al. (2018) [83] Germany	Compulsive exercise	-	-	-CET -EMI-2	4 weeks prior to hospitalization.
Young et al. (2018) [84] Australia	Compulsive exercise	-	-	CET	4 weeks prior to hospitalization.

Acc.: Accelerometer. CEQ: Commitment to Exercise Questionnaire. CES: Commitment to Exercise Scale. CET: Compulsive Exercise Test. CRQ: Commitment to Running Questionnaire. DDE: Dieting Disorder Examination. DFA-Q: Drive for Activity Questionnaire. DLW: Double Labeled Water Method. DSED: Diagnostic Survey of the Eating Disorders. EAI: Exercise Addiction Inventory. EDE: Eating Disorder Examination. EDE-Q: Eating Disorder Examination Questionnaire. EDI-SC: Eating Disorders Inventory-Symptom Checklist. EDS-R: Exercise Dependence Scale—Revised. EEE-C: Eating and Exercise Examination-Computerized. EMA-Q: Ecological Momentary Assessment Questionnaire. EMI-2: Exercise Motivations Inventory-2. EPSQ: The Exercise Participation Screening Questionnaire. GAD: Generalized Anxiety Disorder. GDES: Global Drive to Exercise Score. IPAQ: International Physical Activity Questionnaire. MAQ: Modifiable Activity Questionnaire. METs: Metabolic equivalents [85]. ORQ: Obligatory Running Questionnaire. PA: Physical activity. PAQ: Physical activity questionnaire. PDPAR: Pediatric Physical Activity Recall. REI: Reasons for Exercise Inventory. SIAB: Structured Interview for Anorexic and Bulimic Disorders. SDSS: Substance Dependence Severity Scale. SIAB-EX: Structured Inventory for Anorexic and Bulimic Syndromes. 7d PAR: 7 days Physical Activity Recall. “-”: no available data in study.

**Table 2 nutrients-12-00183-t002:** Status and group classification of studies in the review and prevalence of problematic use of physical activity and statistical analysis used (columns 1 to 3).

1	2	3		
References Country	Group Classification	Prevalence (%)		
		AN	BN	Controls
Falk et al. (1985) [38] USA	Group 1	-	-	-
Kaye et al. (1986) [39] USA	Group 1	-	-	-
Casper et al. (1991) [40] USA	Group 1	-	-	-
Pirke et al. (1991) [41] Germany	Group 1	-	-	-
Long et al. (1993) [42] Canada	Group 2	-	-	-
Brewerton et al. (1995) [44] USA	Group 2 ^b^	38.5	22.5	-
Davis et al. (1995) [45] Canada	Group 1 (Int.)	-	-	Sample 2: 37.5
	Group 2 (CES)			-
Bouten et al. (1996) [46] The Netherlands	Group 1	-	-	-
Casper and Jabine (1996) [47] USA	Group 1	75	-	-
Davis et al. (1997) [48] Canada	Group 2 ^b^	80.8	57.1	-
Davis and Claridge (1998) [49] Canada	Group 1 (Current ex. st.)	47.08	32.65	-
	Group 2 (Lifetime ex. st.)	76.47	57.14	
Davis et al. (1999) [50] Canada	Group 1 (Current ex. st.)	69	-	-
	Group 2 (CES)	-		
Favaro et al. (2000) [51] Italy	Group 1	31.25	-	-
Pinkston et al. (2001) [52] USA	Group 1 (7d PAR)	-	-	-
	Group 2 (CES)			
Solenberger (2001) [3] USA	Group 1 ^b^	54	39	-
Davis and Woodside (2002) [53] Canada	Group 1	-	-	-
Penas-Lledo et al. (2002) [54] Spain	Group 2	46	45.9	-
Holtkamp et al. (2003) [55] Germany	Group 2	-	-	-
Holtkamp et al. (2004) [56] Germany	Group 2	-	-	-
Klein et al. (2004) [57] USA	Group 2	48	-	-
Davis et al. (2005) [58] Canada	Group 1	64	-	2.1 *
Davis and Kaptein (2006) [59] Canada	Group 2	Current ex. st.: 50 Lifetime ex. st.: 80	-	-
Holtkamp et al. (2006) [60] Germany	Group 1 (acc.)	-	-	-
	Group 2 (SIAB)			
Shroff et al. (2006) [61] USA	Group 2 ^b^	ANR: 40.3 ^a,^*; ANP: 54.5 ^a^; ANB: 37.4 ^a,^*; ANBN: 43.5 ^a,^*	BNP: 20.2 ^a^; NPBN: 24 ^a^	-
Klein et al. (2007) [62] USA	Group 1	41.7	-	-
Dalle Grave et al. (2008) [63] Italy	Group 2 ^b^	AN-R: 80; AN-BP: 43.3	39.3	-
Mond and Calogero (2009) [64] Australia	Group 2 ^b^	81.1	-	-
Bewell-Weiss and Carter (2010) [65] Canada	Group 2	34	-	-
Thornton et al. (2011) [66] Sweden and USA	Group 2	AN: 38.7; AN + GAD: 59.1	-	2.5
Carrera et al. (2012) [67] The Netherlands	Group 1	-	-	-
Murray et al. (2012) [68] Australia	Group 2	-	-	-
Smith et al. (2012) [69] USA	Group 2	-	-	-
Alberti et al. (2013) [71] Italy	Group 1	-	-	-
Brownstone et al. (2013) [72] USA	Group 2	-	204	-
El Ghoch et al. (2013) [70] Italy	Group 1	-	-	-
Kostrzewa et al. (2013) [73] The Netherlands	Group 1	30	-	-
Zipfel et al. (2013) [74] Australia	Group 2	58.3	-	16.7 *
Keyes et al. (2015) [75] United Kingdom	Group 1 (act.)	-	-	-
	Group 2 (GDES)			
Sauchelli et al. (2015) [76] Spain	Group 1	37.1	-	61.1
Sternheim et al. (2015) [77] The Netherlands	Group 2	-	-	-
Blachno et al. (2016) [78] Poland	Group 1	-	-	-
El Ghoch et al. (2016) [79] Italy	Group 1	-	-	-
Gianini et al. (2016) [80] USA	Group 1	-	-	-
Noetel et al. (2016) [81] Australia	Group 2	-	-	-
Lehmann et al. (2018) [82] Germany	Group 1	-	-	-
Schlegl et al. (2018) [83] Germany	Group 1 (Self-report)	-	-	-
	Group 2 (CET; EMI-2)			
Young et al. (2018) [84] Australia	Group 2	-	-	-

Acc.: Accelerometer. Act.: Actimetry. AN: Anorexia Nervosa. AN-R: Anorexia nervosa restricting-type. AN-BP: Anorexia nervosa binge-eating/purging type.BN: Bulimia Nervosa. CES: Commitment to Exercise Scale. CET: Compulsive Exercise Test. Current ex. st.: Current exercise status. EMI-2: Exercise Motivation Inventory-2. GAD: Generalized Anxiety Disorder. GDES: Global drive to exercise score. Int.: Interview. Lifetime ex. st.: Lifetime exercise status. Multi: Multivariate statistical analysis. SIAB: Structured Interview for Anorexic and Bulimic Disorders. Uni: Univariate statistical analysis. 7d PAR: 7 days Physical Activity Recall. “-”: No available data in study. * *p* < 0.05. ^a^ According to Shroff and colleagues [61] unique form of categorization of lifetime ED subtypes: RAN: AN with restrictive eating and no purging or bingeing behavior; PAN: AN with purging behavior and no bingeing behavior; BAN: AN with bingeing with or without compensatory behaviors; PBN: BN with purging behavior; NPBN: BN with bingeing and no purging behavior. ^b^ Unfused samples (see “results” paragraph).

**Table 3 nutrients-12-00183-t003:** Methods of studies investigating problematic use of physical activity in ED (columns 1 to 11).

1	2	3	4	5	6	7	8	9	10	11
References Country	Study Design	Diagnostic Criteria	ED Subjects	Controls	Gender	Age (Years) Mean ± SD (Range)	Duration of Illness (Years) Mean ± SD (Range)	Age of Onset of ED (Years) Mean ± SD (Range)	Type of Treatment	Prevalence Period
Falk et al. (1985) [38] USA	Prospective case series	DSM-III	20 AN	-	Women	21.1 ± 5.6	-	-	In	Current
Kaye et al. (1986) [39] USA	Prospective case series	DSM-III	22 AN	11 healthy	Women	RW-R: 25.0 ± 1.1 LtW-R: 24.9 ± 1.7	RW-R = 84.1 ± 13.2 months LtW-R = 100.1 ± 14.7 months	RW-R = 17.6 ± 0.9 LtW-R = 16.5 ± 1.4	-	Lifetime 11 RW-R 11 LtW-R
Casper et al. (1991) [40] USA	Cross-sectional	DSM-III-R	6 AN	6 healthy	Women	24.5 ± 8	22.67 ± 10.7 months	-	Out	Current
Pirke et al. (1991) [41] Germany	Prospective case series	DSM-III-R	8 AN 8 BN	11 healthy	Women	AN: 27.8 ± 5.2 BN: 24.3 ± 4.7	-	-	In ^d^	Current
Long et al. (1993) [42] Canada	Cross-sectional	DSM-III-R	21 AN	62 healthy	AN: women. Controls: 42 women; 20 men	25 ± 9.7	-	-	In	Current
Brewerton et al. (1995) [44] USA	Retrospective case series studies	DSM-III-R	18 AN 71 BN 21 EDNOS	-	Women	-	-	Ces: 12.1 ± 3.0 Non-Ces: 16.3 ± 1.8	In	Current
Davis et al. (1995) [45] Canada	Cross-sectional	DSM-III-R	46 AN	88 regular exercisers 40 high-level exercisers	Women	24.2 ± 4.7	-	-	In	Current
Bouten et al. (1996) [46] The Netherlands	Cross-sectional	DSM-III-R	11 AN	13 healthy	Women	33.6 ± 7.8	5 to 28	-	Out	Current
Casper and Jabine (1996) [47] USA	Follow-up	Feighner criteria	73 AN	-	Women	Early adolescent onset: 16.2 ± 3.3 Late adolescent onset: 19.7 ± 3.3 Adult onset: 25.2 ± 3.7	2.9	Early adolescent onset:13.9 ± 1.1 Late adolescent onset:17.3 ± 0.8 Adult onset: 22.2 ± 2.6	64 In 9 Out	Lifetime
Davis et al. (1997) [48] Canada	Cross-sectional	DSM-IV	Sample 1: 78 AN 49 BN; Sample 2: 40 AN	-	Women	Sample 1: 27.7 ± 7.8 Sample 2: 14.3 ± 1.5	-	-	In and Out	Current
Davis and Claridge (1998) [49] Canada	Cross-sectional	DSM-III-R	34 AN 49 BN	-	Women	28.1 ± 8.2	-	-	In and Out	Current and lifetime
Davis et al. (1999) [50] Canada	Cross-sectional	DSM-IV	84 AN	-	Women	15.36 ± 1.38	11.6 ± 13.3 months	14.39 ± 1.38	In	Current
Favaro et al. (2000) [51] Italy	Cross-sectional	DSM-IV	13 AN-R 3 AN-BP	-	Women	22.3 ± 6.3	32.8 ± 43.4 months	-	Out	Current
Pinkston et al. (2001) [52] USA	Cross-sectional	DSM-III-R	11 EDNOS ^a^	15 healthy	Women	20 ± 1.6	-	-	No treatment	Current
Solenberger (2001) [3] USA	Retrospective case series studies	DSM-IV-R	115 AN 38 BN 46 EDNOS	-	Women	20.6 ± 7.03	4.9 ± 0.49	-	In	Current
Davis and Woodside (2002) [53] Canada	Cross-sectional	DSM-IV	78 AN-R 76 BN 32 EDNOS	-	Women	27.0 ± 8.4	-	-	In and Out	Current
Penas-Lledo et al. (2002) [54] Spain	Retrospective case series	DSM-IV	35 AN-R 28 AN-BP 61 BN	-	Women	AN ex.: 18 ± 5.91 AN non-ex: 20.9 ± 6.38 BN ex: 20.1 ± 3.32 BN non-ex: 21.2 ± 4.72	-	-	Out	Current
Holtkamp et al. (2003) [55] Germany	Cross-sectional	DSM-IV	21 AN-R 6 AN-BP	-	Women	14.54 ± 1.299	-	-	In	Current
Holtkamp et al. (2004) [56] Germany	Cross-sectional	DSM-IV	23 AN-R 7 AN-BP	-	Women	14.6 ± 1	11 ± 7 months	-	In	Current
Klein et al. (2004) [57] USA	Cross-sectional	DSM-IV	8 AN-R 13 AN-BP	-	Women	23.38 ± 4.78	-	-	In	Current
Davis et al. (2005) [58] Canada	Cross-sectional	DSM-IV	125 AN-R 14 AN-BP	94 healthy	Women	15.3 ± 1.4	11.9 ± 1.2 months	14.3 ± 1.5	In	Current
Davis and Kaptein (2006) [59] Canada	Prospective case series	DSM-IV	50 AN-R	-	Women	25.4 ± 9.1	-	-	In	Current and lifetime
Holtkamp et al. (2006) [60] Germany	Cross-sectional	DSM-IV	26 AN	-	Women	15.6 ± 1.9	-	-	In	Current
Shroff et al. (2006) [61] USA	Cross-sectional	DSM-IV	521 RAN ^c^ 336 PAN ^c^ 182 BAN ^c^ 296 PBN ^c^ 25 NPBN ^c^ 400 ANBN 96 EDNOS	-	Women	Ex.: 26 ± 7.68 Non-ex: 27.99 ± 9.51	Ex: 9.27 ± 6.93 Non-ex: 9.41 ± 8.04	-	-	Lifetime
Klein et al. (2007) [62] USA	Cross-sectional	DSM-IV	14 AN-R 22 AN-BP	-	Women	26.3 ± 5.9	-	-	In ^d^	Current
Dalle Grave et al. (2008) [63] Italy	Prospective case series	DSM-IV	35 AN-R 30 AN-BP 28 BNP 72 EDNOS	-	Women	26.0 ± 7.8	Ces: 94.3 ± 92.6 months Non-Ces: 114.4 ± 92.3 months	AN: 16.5 ± 5.2 BN: 17.9 ± 4.3 EDNOS: 17.0 ± 6.0	In ^d^	Current
Mond and Calogero (2009) [64] Australia	Cross-sectional	DSM-IV	15 AN-R 13 AN-BP 41 BN 33 EDNOS	184 healthy	Women	AN-R: 21.0 ± 8.3 AN-BP: 23.6 ± 8.3 BN: 23.8 ± 6.2 EDNOS: 20.5 ± 7.8	-	-	Out	Current
Bewell-Weiss and Carter (2010) [65] Canada	Cross-sectional	DSM-IV	98 AN-R 61 AN-BP	-	148 women 5 men	26.0 ± 8.0	6.4 ± 8.0	-	In ^d^	Current
Thornton et al. (2011) [66] Sweden and USA	Cross-sectional	DSM-IV	32 AN 22 AN+GAD	5424 healthy	Women	AN:34.8 ± 6 AN+GAD: 30.1 ± 6.5	-	-	-	Lifetime
Carrera et al. (2012) [67] The Netherlands	Cross-sectional	DSM-IV	25 AN-R 9 AN-BP 3 EDNOS ^a^	-	Women	15.3 ± 1.25	1.2 ± 0.79	-	5 In 32 Out	Current
Murray et al. (2012) [68] Australia	Cross-sectional	DSM-5	24 AN	15 gym-using	Men	23.92 ± 5.57	-	-	13 In 11 Out	Current
Smith et al. (2012) [69] USA	Cross-sectional	DSM-IV	144 BN 60 EDNOS ^b^	-	Women	25.67 ± 8.85	-	-	-	Current
Alberti et al. (2013) [71] Italy	Prospective case series	DSM-IV	52 AN	-	Women	24.4 ± 8.4	5 ± 9	-	In ^d^	Current
Brownstone et al. (2013) [72] USA	Cross-sectional	DSM-IV	144 BN 60 EDNOS ^b^	-	Women	25.7 ± 8.8	-	-	-	Current
El Ghoch et al. (2013) [70] Italy	Prospective case series	DSM-IV	32 AN-R 21 AN-BP	53 healthy	Women	24.5 ± 8.8	-	-	In ^d^	Current
Kostrzewa et al. (2013) [73] The Netherlands	Prospective case series and follow-up	DSM-IV	25 AN-R 9 AN-BP 3 EDNOS ^a^	-	Women	15.15 ± 1.21	-	-	23 In 14 Out ^d^	Current
Zipfel et al. (2013) [74] Australia	Prospective case series	DSM-IV	8 AN-R 4 AN-BP	12 healthy	Women	21.9 ± 6.2	4.3 ± 3.9	-	In ^d^	Current
Keyes et al. (2015) [75] United Kingdom	Prospective case series	DSM-IV	55 AN	30 healthy 34 anxiety	Women	29	-	-	18 In 37 Out	Current
Sauchelli et al. (2015) [76] Spain	Prospective case series	DSM-IVTR	52 AN-R 36 AN-BP	116 healthy	Women	27.94 ± 9	7.2 ± 6.4	21.2 ± 8.4	In ^d^	Current
Sternheim et al. (2015) [77] The Netherlands	Cross-sectional	DSM-5	145 AN-R 95 AN-BP	-	Women	21.6 ± 8.9	5.5 ± 7.4	15.7 ± 4.0	Out	Current
Blachno et al. (2016) [78] Poland	Prospective case series	DSM-IV and ICD-10	76 AN	-	Women	14.8 ± 1.8	-	-	In	Current
El Ghoch et al. (2016) [79] Italy	Prospective case series and follow-up	DSM-IV	32 AN	-	Women	22.45	-	-	In ^d^	Current
Gianini et al. (2016) [80] USA	Prospective case series and follow-up	DSM-5	61 AN	24 healthy	Women	24.4 ± 6.5	8.3 ± 6.6	-	In	Current
Noetel et al. (2016) [81] Australia	Prospective case series	DSM-5	60 AN	-	Women	15.02 ± 1.22	1.12 ± 0.98	-	In ^d^	Current
Lehmann et al. (2018) [82] Germany	Prospective case series	ICD-10	24 AN-R 13 AN-BP 13 AN atypical	30 healthy	Women	25	6.25	-	In	Current
Schlegl et al. (2018) [83] Germany	Prospective case series	ICD-10	151 AN 75 BN	109 healthy	Women	AN: 21.11 ± 6.94; BN: 23.09 ± 7.6	-	-	In	Current
Young et al. (2018) [84] Australia	Cross-sectional	DSM-5	56 AN-R 22 AN-BP	-	74 women 4 men	27.38 ± 9.22	5.65 ± 7.88	-	Out	Current

AN: Anorexia Nervosa. BN: Bulimia nervosa. DSM: Diagnostic Statistical Manual. EDNOS: Eating disorder not otherwise specified. ED: Eating disorder. AN-R: Anorexia nervosa restricting-type. AN-BP: Anorexia nervosa binge-eating/purging type. AN-BN: lifetime diagnosis of AN and BN. BN-P: Bulimia nervosa purging type. RW-R: Recently weight-recovered; LtW-R: long-term weight-recovered; Ex: Excessive exercisers. Ces: Compulsive exercisers. GAD: Generalized Anxiety Disorder. In: Inpatient treatment. Out: Outpatient treatment. Day: Day treatment. “-”: no available data in study. ^a^ EDNOS patients that were considered AN in review. See review part II. ^b^ EDNOS patients that were considered BN in review. See review part II. ^c^ According to Shroff and colleagues [61] unique form of categorization of lifetime ED subtypes: RAN: AN with restrictive eating and no purging or bingeing behavior; PAN: AN with purging behavior and no bingeing behavior; BAN: AN with bingeing with or without compensatory behaviors; PBN: BN with purging behavior; NPBN: BN with bingeing and no purging behavior. ^d^ Study that gave details about ED treatment program.

**Table 4 nutrients-12-00183-t004:** Instruments used to assess comorbidities, psychological factors, and ED symptomatology (Columns 1 to 4).

1	2	3								4
References Country	Group Classification	Comorbidities and Psychological Factors								ED Symptomatology
		Depression	Anxiety	Ob-Co	Self-Esteem	Stress	Addict.	Regul.	Anhedonia	
Falk et al. (1985) [38] USA	Group 1	HRS	-	-	-	-	-	-	-	-
Casper et al. (1991) [40] USA	Group 1	BDI	-	-	-	-	-	-	-	EDI; EAT
Long et al. (1993) [42] Canada	Group 2	BSI	BSI	-	SEI	-	-	-	-	EAT; EDI
Davis et al. (1995) [45] Canada	Groups 1 and 2	-	-	SCL-90	-	-	-	-	-	EDI
Casper and Jabine (1996) [47] USA	Group 1	-	-	-	GAS	-	-	-	-	-
Davis et al. (1997) [48] Canada	Group 2	-	-	-	-	-	-	-	-	Self-report psychological inventories
Davis et al. (1999) [50] Canada	Groups 1 and 2	-	-	Lazare et al. Inventory *	-	-	EPQ-R	-	-	-
Favaro et al. (2000) [51] Italy	Group 1	HSCL	HSCL	-	-	-	-	-	-	EDI; 24-h dietary recalls; 3-day food record
Davis and Woodside (2002) [53] Canada	Group 1	SCL-90	-	-	-	-	-	-	Physical Anhedonia scale	-
Penas-Lledo et al. (2002) [54] Spain	Group 2	SCL-90-R	SCL-90-R	SCL-90-R	-	-	-	-	-	EAT-40; BITE
Holtkamp et al. (2004) [56] Germany	Group 2	SCL-90-R	SCL-90-R	SCL-90-R	-	-	-	-	-	SIAB
Klein et al. (2004) [57] USA	Group 2	BDI	BAI	-	-	-	-	-	-	YBC-EDS
Davis et al. (2005) [58] Canada	Group 1	-	-	-	-	-	-	-	-	EDI; Interview
Davis and Kaptein (2006) [59] Canada	Group 2	-	-	MOCI; Lazare et al. Inventory *	-	-	-	-	-	-
Klein et al. (2007) [62] USA	Group 1	BDI	BAI	-	-	-	-	-	-	EDI
Bewell-Weiss and Carter (2010) [65] Canada	Group 2	BDI-2	BSI	Padua Inventory	Rosenberg Self-Esteem Scale	-	-	-	-	EDE-Q; EDI
Thornton et al. (2011) [66] Sweden and USA	Group 2	-	GAD	-	-	-	-	-	-	-
Carrera et al. (2012) [67] The Netherlands	Group 1	CDI	STAI	-	-	-	-	-	-	EDI-2
Brownstone et al. (2013) [72] USA	Group 2	-	-	-	-	-	-	DAPP-BQ	-	EDE-Q
Kostrzewa et al. (2013) [73] The Netherlands	Group 1	CPRS-S-A	CPRS-S-A	CPRS-S-A	-	-	-	-	-	EDI-2; MROAS
Zipfel et al. (2013) [74] Australia	Group 2	BDI	-	-	-	-	-	-	-	EAT; EDI-2
Keyes et al. (2015) [75] United Kingdom	Groups 1 and 2	DASS	DASS	-	-	DASS	-	-	-	EDE-Q
Sauchelli et al. (2015) [76] Spain	Group 1	SCL-90-R	-	-	-	-	-	-	-	EDI-2
Sternheim et al. (2015) [77] The Netherlands	Group 2	-	STAI	-	-	-	-	-	-	EDE
Blachno et al. (2016) [78] Poland	Group 1	-	-	LOI-CV	-	-	-	-	-	-
Noetel et al. (2016) [81] Australia	Group 2	RCADS	RCADS	ChOCI-R	Rosenberg Self-Esteem Scale	-	-	-	-	Y-EDEQ

Addict.: Addictiveness. ED: Eating disorders. BAI: Beck Anxiety Inventory. BSI: Brief Symptom Inventory. BITE: Bulimic Investigatory Test, Edinburg. ChOCI-R: Children’s Obsessional Compulsive Inventory-Revised. CPRS-S-A: Comprehensive Psychopathological Rating Scale. DAPP-BQ: Dimensional Assessment of Personality Pathology - Basic Questionnaire. DASS: Depression Anxiety Stress Scale 21-version. EAT: Eating Attitudes Test. EDE-Q: Eating Disorder Examination–Self-Report Questionnaire Version. EDI: Eating Disorder Inventory. EPQ-R: Eysenck Personality Questionnaire-Revised. GAD diagnosis: Generalized anxiety disorder diagnosis. GAS: Global Assessment Scale. HRS: Hamilton Rating Scale. HSCL: Hopkins Symptom Checklist. LOI-CV: Leyton Obsessional Inventory-Child Version. MMPI: Minnesota Multiphasic Personality Inventory. MOCI: Maudsley Obsessive–Compulsive Inventory. MROAS: Morgan and Russell Outcome Assessment Schedule. Ob-Co: Obsessive–Compulsiveness. RCADS: Revised Child Anxiety and Depression Scale. Regul.: Regulation and verbal expression of emotions. SCID: Structured Clinical Interview for DSM-IV. SCL-90: Symptom Check-List-90. SCL-90-R: Symptom Check-List-90-Revised. SEI: Culture Free Self Esteem Inventory. SIAB: Structured Interview of Anorexia and Bulimia Nervosa. STAI: State-Trait Anxiety Inventory. YBC-EDS: Yale-Brown-Cornell Eating Disorder Scale. Y-EDEQ: Youth Eating Disorder Examination-Questionnaire. “-”: No available data in study: * Inventory designed to assess the “obsessional” or “anal” personality type derived from psychoanalytic theory [86,87].

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
