# Peer review of "Physical Activity in Eating Disorders: A Systematic Review"

_nutrients, 2020, doi:10.3390/nu12010183_

Round 1

Reviewer 1 Report

The article under evaluation provides a proposed suggestion to conceptualize increased physical activity as “problematic PA” in patients with eating disorders. Based on a systematic literature review, 2 categories and groups are proposed, one referring to quantitative excessive exercise and on to pathological motivation to exercise. Clinical correlates in theses groups as well as a model for dynamic development of different PA phases and forms are presented. The topic is of high importance, the article is comprehensive, well researched and written, and provides new, interesting and relevant aspects and information on the topic of physical activity in eating disorders. However, some points remain unclear.

Major concerns:

While I appreciate the effort of finding a common term for the phenomenon of increased PA, I am not sure about the proposed solution. “Problematic” suggests that there is a problem, yet it not described what kind of problem this is, and for whom (patients or carers) or what (weight gain or physical or psychological health) the problem exists. As the review itself mentions, increased PA also has positive aspects, such as mood regulation or as a coping strategy. Thus, in this regard, PA might not be problematic but an intent by the patients so solve the problem of anxiety. Problematic in this situation could also be that PA in ED is not well understood, that it is restricted and not therapeutically addressed, and thus needs to be carried out in secret on inpatient wards to avoid restriction. In summary: how do the authors explain the choice of the word “problematic”? How does this term add, or is superior to, other suggested terms (i.e. the term of compulsive exercise proposed by Dittmer et al “Compulsive exercise in eating disorders: proposal for a definition and a clinical assessment”, J Eat Disord 2018?; Davis 2015: “compulsive activity” or “compulsive exercise” proposed by Noetel et al. in 2016? How is the fact accounted for in the proposed term that – depending on the nutritional status of an individual patient “PPA” could be problematic for one patient, and not for another? The separation into quantitative and qualitative features seems important and logical, yet could there be more neutral term than problematic and pathological? It seems that a range of research groups around the globe keep suggesting new terms and concepts with respect to describing increased PA in AN, and that these groups keep repeating and adhering to their own suggestions, so that a common terminology is not in sight. On the contrary, every new suggestion by a single group increases the plethora of terms in use. I would strongly suggest to include this issue in the review, and to come up with a solution, namely to build an international and interdisciplinary task force, to have the task force work on and come to such an urgently needed commonly accepted solution, that is then, in a second step, incorporated in guidelines for further research in the field. To put it in different words: what is the author’s suggestion in order to harmonize existing and future and new concepts of the PA terminology? The setting (inpatient versus outpatient), as well as the rules in this setting, strongly affects PA. The authors mention this aspect in the discussion. However, how is this fact accounted for in the results section as well as in the proposed model of dynamic PA phases? Methods: the few search terms “Exercise” OR “physical activity” seem restrictive. Why did the authors not expand with further relevant terms, such as “hyperactivity” or “restlessness” "motor drive"? How do they estimate the effect of the restrictive search terms on the results? In the some parts of the discussion PPA is differentiated in EEC and PME, in others this separation is not done. What is the reason for this discrepancy? the model of the development of PA seems intriguing and interesting, yet also quite subjective. It is not clear, how the timing was derived. How can we be sure that certain ED/AN subgroups would not display the phases in a different order? The article lacks a limitation section The conclusion is not clear. In the paper, the aspects “excessive exercise” and “pathological motivation” are addressed first and systematically; then, in the subjective/narrative model, the factors “voluntary” and “involuntary” are added, However in the conclusion it seems like these 4 terms and concepts are mixed. Does the paper support the PPA term, and why?

Minor comments

- Abstract: line 39 „components“ instead of „component”

- Introduction line 51: what is meant with “partial”

- throughout the manuscript, there are a few spelling mistakes and the layout (e.g. using bold script for headers, Capital letter at the beginning) is not entirely consistent

- Flowchart: why are some articles mentioned in legend and others not, what is meant with part 2?

- Table 2: how does the choice of statistics effect the question and why is this mentioned in the table?

- line 167: please explain (20/20) in the header

- line 170: is the number 12 placed accidentally?

Author Response

Reviewer #1:

Major concerns:

While I appreciate the effort of finding a common term for the phenomenon of increased PA, I am not sure about the proposed solution. “Problematic” suggests that there is a problem, yet it not described what kind of problem this is, and for whom (patients or carers) or what (weight gain or physical or psychological health) the problem exists. As the review itself mentions, increased PA also has positive aspects, such as mood regulation or as a coping strategy. Thus, in this regard, PA might not be problematic but an intent by the patients so solve the problem of anxiety. Problematic in this situation could also be that PA in ED is not well understood, that it is restricted and not therapeutically addressed, and thus needs to be carried out in secret on inpatient wards to avoid restriction.

In summary: how do the authors explain the choice of the word “problematic”? How does this term add, or is superior to, other suggested terms (i.e. the term of compulsive exercise proposed by Dittmer et al “Compulsive exercise in eating disorders: proposal for a definition and a clinical assessment”, J Eat Disord 2018?; Davis 2015: “compulsive activity” or “compulsive exercise” proposed by Noetel et al. in 2016?

How is the fact accounted for in the proposed term that – depending on the nutritional status of an individual patient “PPA” could be problematic for one patient, and not for another?

The separation into quantitative and qualitative features seems important and logical, yet could there be more neutral term than problematic and pathological?

It seems that a range of research groups around the globe keep suggesting new terms and concepts with respect to describing increased PA in AN, and that these groups keep repeating and adhering to their own suggestions, so that a common terminology is not in sight. On the contrary, every new suggestion by a single group increases the plethora of terms in use. I would strongly suggest to include this issue in the review, and to come up with a solution, namely to build an international and interdisciplinary task force, to have the task force work on and come to such an urgently needed commonly accepted solution, that is then, in a second step, incorporated in guidelines for further research in the field.

To put it in different words: what is the author’s suggestion in order to harmonize existing and future and new concepts of the PA terminology? The setting (inpatient versus outpatient), as well as the rules in this setting, strongly affects PA. The authors mention this aspect in the discussion. However, how is this fact accounted for in the results section as well as in the proposed model of dynamic PA phases?

In the some parts of the discussion PPA is differentiated in EEC and PME, in others this separation is not done. What is the reason for this discrepancy?

In the paper, the aspects “excessive exercise” and “pathological motivation” are addressed first and systematically; then, in the subjective/narrative model, the factors “voluntary” and “involuntary” are added, However in the conclusion it seems like these 4 terms and concepts are mixed. Does the paper support the PPA term, and why?

Response of the Authors: Thank you for your remarks concerning the terms to be used to describe physical activity (PA) in eating disorders (ED).

Before writing this review, we had given this matter consequent attention and thoughts.

Indeed, as you highlighted, there is already a large plethora of terms in the literature; why add more to it and make it more complicated than it already is? However, after completing a thorough review of the literature, it seemed that all the terms that we found only described one dimension of PA in ED. We finally decided to stick with the term “problematic physical activity” (PPA) for many reasons:

Why “Physical activity”?

Perhaps the most well known and most cited definition of PA comes from Caspersen, Powell and Christenson (1985), who describe it as “any bodily movement produced by skeletal muscles that results in energy expenditure” (p. 126). This definition includes the three subgroups of physical (exercise, daily living activities and hobbies). It also includes any bodily movement noticed in ED patients such as fidgeting, motor restlessness, standing on one leg, etc. This being said, a lot of terms that only used “exercise” automatically omitted the other subgroups of PA mentioned above.

When reviewing all the terms found in the literature, it seemed they did not take into account both the qualitative and quantitative dimension of PA in ED. Terms like “compulsive exercise” used by many teams (Noetel et al. (2016), Dalle Grave et al. (2008) (kindly check Table 1 column 2 for further references) or “drive for exercise” by Sternheim et al. (2015) only take into account the qualitative dimension, hence the motivation to exercise, without taking into account the 3 characteristics of exercise (Duration, Frequency and Intensity). In addition, terms such as “compulsive exercise” also only taking into account one subgroup of PA. Exercise is a subgroup of PA that is planned, structured, repetitive, and purposeful (World Health Organization, 2005). Thus, using the term “exercise” omits the rest of the subgroups (daily living activities, hobbies, motor restlessness, etc.). Now we have found a very interesting term in the literature: “unhealthy exercise” (Noetel et al., 2017 and Touyz et al., 2017). However, we fall back into the issue of omitting other subgroups of PA by using the word “exercise” (this term was added in our review, line 69).

Why “problematic”?

PA is problematic both for the person and the treatment: it can lead to injuries, it upsets the person as it is compulsive. It also interferes with nutrition rehabilitation.

This review was preceded by a research that we have done on testing many definitions of exercise found in ED (Rizk M, Lalanne C, Berthoz S, Kern L, EVHAN Group, Godart N (2015) Problematic Exercise in Anorexia Nervosa: Testing Potential Risk Factors against Different Definitions. PLoS ONE 10 (11): e0143352). This article and our previous works accentuated the fact problematic physical activity had a serious negative influence on ED patients during the whole evaluation of the disorder. It is a core symptom of ED and a restrictive behaviour used by patients to lose weight (APA, 2013). It also interferes with nutrition rehabilitation and increases the risk of short-term somatic complications such as fractures and bruises (Rizk, Kern, Godart, & Melchior, 2014). It is associated with worse clinical and therapeutic outcomes (El Ghoch et al., 2013; Ng, Ng, & Wong, 2013; Solenberger, 2001). Thus, problematic physical activity plays a crucial role in the lives of many ED patients, interferes in their treatment outcome and in the efficiency of their nutritional rehabilitation. Consequently, problematic physical activity could have a worst impact on an individual suffering from an ED, according to his/her nutritional status. And, we also agree that the setting (inpatient versus outpatient) strongly affects PA. Therefore, we thoroughly analyzed the setting of each targeted population in the studies included in the review (Table 3, column 10) and have divided our results as well as our discussion accordingly. 

We strongly agree with you that problematic physical activity seems to affect quality of life of patients, but not always in a negative way, as you have mentioned in your comment. Essentially, our work highlights that physical activity should perhaps not only be viewed in a negative way as previously described. It surely should be targeted in global treatment programs in addition to the psychological, somatic and social dimensions of treatment.  

As you proposed, we strongly agree that problematic physical activity in ED is in need of immediate attention from clinicians and researchers. Research on introducing physical activity in the treatment of ED are needed. There is a recent growing interest in the place exercise could have as a component in the treatment of several mental disorders (Zschucke, Gaudlitz, & Strohle, 2013), including eating disorders (Hausenblas, Cook, & Chittester, 2008; Vancampfort et al., 2014) and AN (Ng et al., 2013; Zunker, Mitchell, & Wonderlich, 2011). In fact, with the help of fundamental research on animal models, including pharmacological and clinical researches, a clear clinical consensual definition of PPA and its different dimensions can hopefully open the path to new and developed treatment perspectives.

This being said, we strongly agree that there is an urgent need for the development of an international and interdisciplinary task force that would urgently need to find a commonly accepted term, definition and solution to the issue. We have added this part to the review (lines 596 and 626).

Furthermore, in the light of your successive comments related to the use of several terminology, we agree that we need to adopt a minimalistic approach to our terminology rather than confusing the readers more. To simplify our review, we have completely removed the terms EEC (excessive exercise characteristics) and PME (pathologically motivated exercise) and only adopted the groups’ numerations (group 1 & group 2). 

Finally, we are aware that our team has specifically chosen the term “problematic”. However, we have failed to find a more neutral term that would encompass both problematic dimensions of physical activity in ED. Following your enriching comment, we have added this fact in the limitation section (line 594).

You have mentioned that PA is used by individuals with AN both in a positive way (in order to alleviate anxiety and depressive symptoms) and in a negative way (to lose weight, in a compulsive way…). It is actually the same for benzodiazepine that can be used positively in order to alleviate anxiety, but can lead to an addiction use that is problematic.

Taking into account all the above and the fact that PA is not problematic per se but becomes problematic due to its use, we propose to modify “problematic physical activity” and replace it with “problematic use of physical activity”. 

Methods: the few search terms “Exercise” OR “physical activity” seem restrictive. Why did the authors not expand with further relevant terms, such as “hyperactivity” or “restlessness” "motor drive"? How do they estimate the effect of the restrictive search terms on the results?

Response of the Authors: Thank you for highlighting this issue. We focused on the two terms “exercise” OR “physical activity” mainly because these two terms, especially “physical activity” automatically encompasses all terms related to any muscle contraction produced by skeletal muscles that requires energy expenditure (Caspersen, Powell and Christenson, 1985). Hence, any search including “physical activity” will automatically include terms such as hyperactivity, restlessness, motor drive and even inner drive. In addition, our manual research while reading papers did not lead us to discover articles that where not found by our research.  

the model of the development of PA seems intriguing and interesting, yet also quite subjective. It is not clear, how the timing was derived. How can we be sure that certain ED/AN subgroups would not display the phases in a different order?

Response of the Authors: We included the elements of the literature in the model according to a time line defined by the nature of the elements. For example, at the beginning, we mentioned risk factors described in the literature: the father’s PA status, and PA during childhood. Progressively then, we included the influence of the ambient temperature on PA or biological elements linked to malnutrition (animal models). This is a hypothetical model based on clinical experience, the testimony of individuals with AN, and the process of learning motor activity (Perruchet & Gallego, 1997; Steinglass & Walsh, 2006).

We agree that it does not concern all the persons suffering from AN since some do not have PPA. We added this in the limitations section (line 601). 

The article lacks a limitation section

Response of the Authors: We have added a limitations section (line 593).

The conclusion is not clear.

Response of the Authors: We reformulate the conclusion in order to clarify it. 

Minor comments

- Abstract: line 39 „components “instead of „component”

Response of the Authors: Corrected (line 40).

- Introduction line 51: what is meant with “partial”

Response of the Authors: We meant by “partial”: non-exhaustive. If this is not clear, we would be willing to change the word “partial” to “non-exhaustive” in the manuscript. 

- throughout the manuscript, there are a few spelling mistakes and the layout (e.g. using bold script for headers, Capital letter at the beginning) is not entirely consistent

Response of the Authors: Thank you for this remark. We have done the corrections accordingly in the manuscript.  

- Flowchart: why are some articles mentioned in legend and others not, what is meant with part 2?

Response of the Authors: Thank you for highlighting this typography mistake. We have corrected the flowchart and removed “part 2”.

The articles mentioned in the legend are not clear in the version of the manuscript you have received. The correct legend is the below: 

aBlumenthal et al. (1984); Crisp et al. (1980) ; Davis et al. (1994) ; Frey et al. (2000); Kron et al. (1978); Monell et al. (2018) ; Sharp et al. (1994).

bBlumenthal et al. (1984); Higgins et al. (2013); Long and Hollin (1995).

cBoyd et al. (2007); Bratland-Sanda et al. (2010a , 2010b & 2011); Carruth and Skinner (2000); Davis et al. (1994 & 1998); Hechler et al. (2008); Naylor et al. (2011); Stiles-Shields et al. (2011) ; Vansteelandt et al. (2007). 

- Table 2: how does the choice of statistics effect the question and why is this mentioned in the table?

Response of the Authors: In the light of your comment, we have removed the choice of statistics in Table 2. 

- line 167: please explain (20/20) in the header

Response of the Authors: (20/47) in line 183 is translated by the number of studies that have found a prevalence of problematic physical activity. This means that 20 studies out of the 47 studies included in this review have mentioned a prevalence of problematic use of physical activity in their results.

- line 170: is the number 12 placed accidentally?

Response of the Authors: Thank you for highlighting this mistake. The number 12 was corrected with 14 (line 187). The corrected sentence is: “In group 2, we found 14 prevalence for PPA in 12 studies”.

14 prevalence: 10 prevalence of current AN (Brewerton et al. (1995); Davis et al. (1997); Davis and Claridge (1998); Penas-Lledo et al. (2002); Klein et al. (2004) ; Davis and Kaptein (2006); Dalle Grave et al. (2008) ; Mond and Calogero (2009); Bewell-Weiss and Carter (2010); Zipfel et al. (2013)); 4 prevalence of lifetime AN (Davis and Claridge (1998); Davis and Kaptein (2006); Shroff et al. (2006); Thornton et al. (2011)).

Reviewer 2 Report

Rizk and colleagues provided a systematic review of the influence of physical activity in eating disorders. The overall hypothesis, based on the currrent literature, is that the problematic physical activity can be classified as excessive exercise (EEC) or pathologically motivated exercise (PME). Overall, this is an informative systematic review and will contribute to further classification and potential treatment of anorexia nervosa (AN) and bulimia nervosa. There are some concerns that need the authors attentions.

The introduction is rather brief and does not explain the important of adhere to the recommendation of Hagger (2012)?  The introduction should include why it is important to re-classify problematic physical activity.  

Classification of EEC and PME are based on the primary literature that met the criteria for inclusion. Because the information to the RizK and colleagues is limited,there is an external bias. The bias is not discussed. That is, the classification of EEC was based on the primary authors quantitative approaches, whereas PME was based on qualitative approaches. Rizk and colleagues interpretations are based on how the data was initially collected.Rizk and colleagues should further discuss (Davis et al, 1995; Davis Claridge, 1998; Holtkamp et al, 2006; Keyes et al, 2005; Schlegl et al., 2018), whick classified both EEC and PME, in their discussion of potential bias.   

Age of a previous study is not a strong rationale for starting a new study. The implication is that old is obsolete. The authors use the term "old" twice (line 63/line 96). Rizk and colleagues should state why the "old" studies are not satisfactory for the current field. What is old? The authors include studies from 1985 (Falk et al), that's 35 years old. Is that not considered "old"? 

Table 2 is jumbled.

Author Response

Reviewer #2:

Rizk and colleagues provided a systematic review of the influence of physical activity in eating disorders. The overall hypothesis, based on the current literature, is that the problematic physical activity can be classified as excessive exercise (EEC) or pathologically motivated exercise (PME). Overall, this is an informative systematic review and will contribute to further classification and potential treatment of anorexia nervosa (AN) and bulimia nervosa. 

Response of the Authors: Thank you for finding our study of interest.

There are some concerns that need the authors attentions.

The introduction is rather brief and does not explain the important of adhere to the recommendation of Hagger (2012)?

Response of the Authors: Thank you for highlighting this issue. We have tried to clarify this point in the introduction (line 64).

The introduction should include why it is important to re-classify problematic physical activity.

Response of the Authors: Thank you for your remark. We have moved a paragraph that clearly states the importance of re-classifying problematic physical activity from the results section to the introduction (line 68).

Classification of EEC and PME are based on the primary literature that met the criteria for inclusion. Because the information to the RizK and colleagues is limited, there is an external bias. The bias is not discussed. That is, the classification of EEC was based on the primary authors’ quantitative approaches, whereas PME was based on qualitative approaches. Rizk and colleagues interpretations are based on how the data was initially collected. Rizk and colleagues should further discuss (Davis et al, 1995; Davis Claridge, 1998; Holtkamp et al, 2006; Keyes et al, 2005; Schlegl et al., 2018), which classified both EEC and PME, in their discussion of potential bias.

Response of the Authors: Thank you for your comment. Actually, each study was classified in group 1 or group 2 depending on the instrument it used to assess physical activity (kindly refer to Table 2, column). If the instrument assessed physical activity using a quantitative approach in terms of intensity, frequency, duration and/or type of physical activity, the study was classified in group 1. If the instrument assessed physical activity in a qualitative approach by focusing on the relation that links an individual to his/her physical activity, including motives for exercise, compulsivity and exercise dependence/addiction, then the study was classified in group 2. Now the studies mentioned in the comment above (Davis et al, 1995; Davis Claridge, 1998; Holtkamp et al, 2006; Keyes et al, 2005; Schlegl et al., 2018) have assessed physical activity both quantitatively and qualitatively. Their results were divided according to which instrument was used. We understand however that this could be a source of confusion and bias and we have included this issue in the limitations section (line 598).

Age of a previous study is not a strong rationale for starting a new study. The implication is that old is obsolete. The authors use the term "old" twice (line 63/line 96). Rizk and colleagues should state why the "old" studies are not satisfactory for the current field. What is old? The authors include studies from 1985 (Falk et al), that's 35 years old. Is that not considered "old"?

Response of the Authors: Thank you for this remark. We agree that the used of the word “old” is inappropriate and we have replaced it with “have been published earlier” (line 66).

Table 2 is jumbled.

Response of the Authors: Thank you for your comment. We agree that table 2 is bulky, and we removed the column on statistical analysis. Please let us know if you have other suggestions to improve it.

Reviewer 3 Report

On the light of the heterogeneity  of results and definition in previous literature studies on this topic,  the authors' effort to provide a systematic review on "physical activity" in eating disorders is appreciable. Introduction, aims, materials and methods are clearly described.

Results concerning the study aims 1 and  2 are well presented, however, it could be better to describe also the results of the study  aim 3 in the Results section. Indeed, this point appears in the Discussion section (paragraph 4.9) without any sequential link with Results. Moreover, the discussion seems a bit ripetitive of the results, and the interpretation of observed data does not seem particularly innovative and could be further developed. For instance, it would be interesting to go into more detail concerning differences with previous review on this topic and the practical implications of present findings. Some of the adopted terms should be replaced by more appropriate ones: for example "psychological factors" is too generic to define depression and anxiety and "ED symptomatology" instead of weight preoccupation or drive for thinness seems far from psychopathology.

In conclusion, please consider reorganizing Results and Discussion, it would bring greater value and clarity to this review   

Author Response

Reviewer #3:

On the light of the heterogeneity of results and definition in previous literature studies on this topic, the authors' effort to provide a systematic review on "physical activity" in eating disorders is appreciable. Introduction, aims, materials and methods are clearly described.

Response of the Authors: Thank you for finding our study of interest.

Results concerning the study aims 1 and 2 are well presented, however, it could be better to describe also the results of the study aim 3 in the Results section. Indeed, this point appears in the Discussion section (paragraph 4.9) without any sequential link with Results.

Response of the Authors: Thank you for your comment. Actually, the basis of aim 1 was to set the foundation of our classification. Aim 2 stem from the results of aim 1. And finally, aim 3 (which is the comprehensive model) was set in the discussion since it’s the result of both aims 1 and 2. However, we can move aim 3 to the results sections if it seems more plausible to you.

Moreover, the discussion seems a bit repetitive of the results, and the interpretation of observed data does not seem particularly innovative and could be further developed. For instance, it would be interesting to go into more detail concerning differences with previous review on this topic and the practical implications of present findings.

Response of the Authors: Thank you for highlighting the redundancy. We made sure to remove sentences re-stating the results and attacking directly on the discussion (lines 431 & 446).

 Some of the adopted terms should be replaced by more appropriate ones: for example "psychological factors" is too generic to define depression and anxiety and "ED symptomatology" instead of weight preoccupation or drive for thinness seems far from psychopathology.

Response of the Authors: Thank you for this comment. In the light of it, we have changed the subtitles from “psychological factors” to “comorbidities and psychological factors” (lines 231 and 233).

In conclusion, please consider reorganizing Results and Discussion, it would bring greater value and clarity to this review  

Response of the Authors: Thank you for your comment. We have revised the Results and Discussion sections according to your previous comments and made them clearer to read.

Round 2

Reviewer 3 Report

The authors addressed adequately all my concerns. I have no further comments.